# Why East Asian monsoon anomalies are more robust in post El Niño than in post La Niña summers

Pengcheng Zhang [1], Shang-Ping Xie [1] ✉, Yu Kosaka [2], Nicholas J. Lutsko [1], Yuko M. Okumura[3] & Ayumu Miyamoto [1]

The East Asian summer monsoon (EASM) supplies vital rainfall for over one billion people. El Niño-Southern Oscillation (ENSO) markedly affects the EASM, but its impacts are more robust following El Niño than La Niña. Here, we show that this asymmetry arises from the asymmetry in ENSO evolution: though most El Niño events last for one year, La Niña events often persist for 2-3 years. In the summers between consecutive La Niña events, the concurrent La Niña opposes the delayed effect of the preceding winter La Niña on the EASM, causing a reduction in the magnitude and coherence of climate anomalies. Results from a large ensemble climate model experiment corroborate and strengthen the observational analysis with an order of magnitude increase in sample size. The apparent asymmetry in the impacts of the ENSO on the EASM can be reduced by considering the concurrent ENSO, in addition to the ENSO state in the preceding winter. This has important implications for seasonal climate forecasts.

In the summer of 2022, an unprecedented heatwave struck a large area of south China and set new temperature records for 361 weather stations, causing heavy socioeconomic losses[1,2]. Apart from anthropogenic forcing that increases the intensity and frequency of extreme heat events in general[3], a low-level anomalous anticyclone (AAC, Fig. 1a) centered over the Northwestern Pacific (NWP) played a crucial role in creating the meteorological conditions favorable for the intensity and persistence of this heatwave[4–6].

The AAC is a recurrent pattern of EASM interannual variability (Fig. 1b)[7–9]. As an intrinsic mode that arises from mean flow confluence and ocean-atmosphere coupling, an AAC is usually triggered by El Niño in the preceding winter, with the Indian Ocean (IO) acting as an intermediary through a capacitor effect[10–14]. In fact, the preceding ENSO explains a large portion of the variability of the summer AAC, resulting in a robust lagged correlation (Fig. S1)[12]. However, this relationship does not guarantee a deterministic progression; for instance, although an anomalous cyclone (AC, or negative AAC) was probabilistically favored to follow the La Niña in the winter of 2021/2022, the

observed outcome was a strong AAC responsible for the record-breaking heatwave.

Looking back at historical observations, the case of 2022 is not the only exception. Year-by-year analysis indicates that both AACs and ACs are observed in post-La Niña summers, whereas post-El Niño summers prefer positive AAC events (Fig. S2). The lagged correlation between preceding ENSO and summer AAC events is largely the result of "winter El Niño to summer AAC" events, and the opposite − i.e., an AC following a La Niña event − is less robust (e.g., see Fig. 2a). The anomalous precipitation in summer, influenced by vapor transport and wind convergence/divergence associated with the AAC, is thus less predictable in post-La Niña summers, impeding the ability of early warning for hazards prevention[15,16].

ENSO itself is not symmetric: in historical records, consecutive La Niña events are more likely to occur than consecutive El Niño events[17–21]. Therefore, it is possible that consecutive La Niña events have a different effect on the summer AAC than single-year ENSO events[4,22,23]. Nevertheless, our understanding of asymmetric AAC

[1]Scripps Institution of Oceanography, University of California San Diego, La Jolla, CA, USA. [2]Research Center for Advanced Science and Technology, The University of Tokyo, Tokyo, Japan. [3]Institute for Geophysics, Jackson School of Geosciences, The University of Texas at Austin, Austin, TX, USA. ✉e-mail: sxie@ucsd.edu

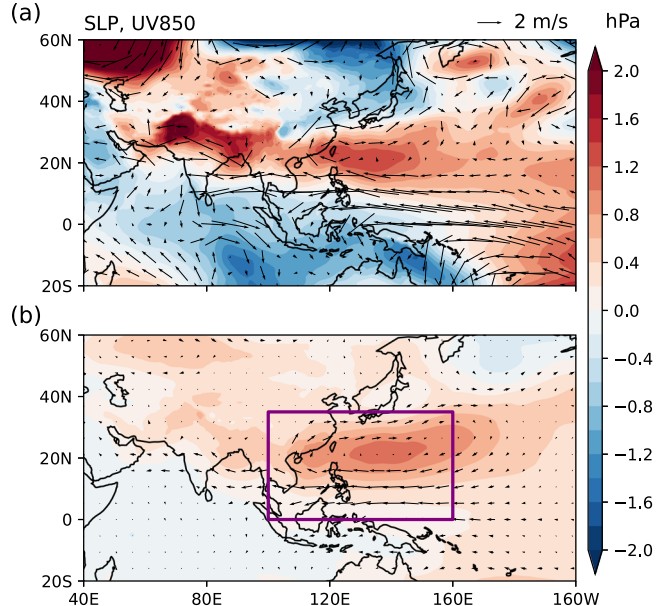

**Fig. 1 | Anomalous anticyclone (AAC) over the Northwestern Pacific (NWP). a** Anomalous sea level pressure (SLP, shading) and 850 hPa winds (arrows) in June-August (JJA) of 2022. **b** Anomalous JJA SLP (shading) and 850 hPa wind (arrows) regressed against the summer AAC index, defined as the leading principal component of the Empirical Orthogonal Function (EOF) analysis on JJA SLP anomalies in the purple box. Source data are provided as a Source Data file.

responses to ENSO is limited by the short duration of reliable atmospheric observations[24]. Furthermore, the leading mode of non-ENSO variability also projects onto the AAC and can emerge spontaneously, making it difficult to extract a clear ENSO-AAC relationship[25–28].

Combining reanalysis data and coupled general circulation model (CGCM) ensemble simulations constrained by realistic ENSO cycles, here we provide a robust and comprehensive explanation for the asymmetry between post-El Niño and post-La Niña summers. We show that La Niña which survives through to the following summer contributes to an AAC rather than an AC, and that if the tropical Pacific cooling is strong enough this contribution wins out over the capacitor effect, as was the case in 2022. By incorporating the summer ENSO phase, we can effectively account for the asymmetry in the observed ENSO-AAC relationship.

## Results
### Asymmetry in post-ENSO summers
We introduce an index to represent the interannual variability of AAC based on the empirical orthogonal function (EOF) analysis of summer sea level pressure (SLP) anomalies (see details in the "Methods" section). The correlation between the summer AAC index and the preceding winter ENSO, indicated by the November–January (NDJ) sea surface temperature (SST) anomalies in Niño 3.4 region, reaches 0.41 during the satellite era of 1979–2022 (Fig. S1). However, behind the overall correlation lies an asymmetry between ENSO phases: the average AAC index in post-El Niño summers is 0.75, but in post-La Niña summers the average AAC index is only −0.27 (Fig. S2; see also the composite analysis in Fig. S3). Despite the small sample size, this difference in the magnitudes of the correlations is close to the significance threshold ($p = 0.076$) using Welch's $t$-test, suggesting the need for further exploration. A close inspection of the scatter plot reveals that the AAC events in post-La Niña summers (dots in quadrants III and IV) do not contribute much to the positive correlation, whereas 73% of post-El Niño events are confined to quadrant I (Fig. 2a). By selecting the summers between consecutive La Niña events (red dots in Fig. 2a), we find that all but one deviate from and fall below the

diagonal, resulting in large asymmetry in the scatter plot. The composite analysis based on observational data reveals that the summer monsoon anomalies between consecutive La Niña events do not exhibit a typical negative AAC pattern, in contrast to those observed in summers following single-year La Niña events (Fig. S4). Therefore, consecutive La Niña events emerge as a key factor in resolving the asymmetry.

La Niña tends to decay more slowly than El Niño[18,29–31]. In consecutive La Niña events, cold anomalies in the tropical Pacific can persist throughout the year despite SST anomalies weakening between the two peak winters. The cold SST anomalies in the tropical Pacific contribute to the AAC after a fast decaying El Niño[32–34]; here we show that similar SST anomalies originating from lingering La Niña events cause the asymmetric post-ENSO AAC responses. We use a multiple linear regression model to consider the NDJ Niño 3.4 SST anomalies, as well as June–August (JJA) SST anomalies in the Niño 4 region to track the concurrent ENSO (see details in "Methods" section; the partial regression coefficients are both significant at 1% using a Student's $t$-test). We then compare the reconstructed AAC index with that from raw reanalysis (Fig. 2b). The Eastern Pacific cold tongue is strong during JJA, and moderate SST perturbations in the Niño 3 region have little atmospheric effect[35,36]. For this reason, we use SST anomalies in the Niño 4 region rather than the Niño 3.4 region for the concurrent ENSO (especially La Niña). We find that, although the concurrent correlation between ENSO and AAC is marginal in JJA (Fig. S1), more AAC variability in post-La Niña summers can be explained by incorporating summer ENSO, as the dots originally in quadrant IV are pushed toward the diagonal (for example, the red dots in Fig. 2b representing persistent cold anomalies in the tropical Pacific associated with multi-year La Niña events). Consequently, the correlation between the reconstructed and observed AAC indices is higher (from 0.41 to 0.54 during the same period). Though this change is not statistically significant due to the small sample size ($p = 0.23$), the asymmetry in the scatter distribution is alleviated (compare Fig. 2b with 2a, especially the red dots).

Although reliable SST observations date back to the 1950s[37], high-quality atmospheric data with global coverage only became available during the satellite era that nominally started in 1979. This restriction on the analysis period limits our ability to make robust inferences about the ENSO-AAC relationship, particularly as it reduces the number of multi-year La Niña events available. To address this limitation, we perform 20-member ensemble simulations with a CGCM, the Geophysical Fluid Dynamics Laboratory Coupled Model version 2.1, which is constrained by observations (see details in "Methods" section). Initialized on January 1st each year from a 20-member pacemaker experiment where tropical Pacific SST anomalies are restored to observations, the CGCM is then run freely for another year. The CGCM experiments create 20 different outcomes for summer ENSO (e.g., see Fig. S7 for simulated ENSO evolution in 1999 and 2022) and other climate conditions while sharing nearly identical ENSO states in the preceding winters that closely resemble observations. Our large-ensemble CGCM experiments increase the number of "between La Niña" summers from 8 in observations to 95, a 12-fold increase that permits rigorous testing of our hypothesis.

The CGCM experiments confirm the existence of the asymmetry between post-El Niño and post-La Niña summers (Fig. 2c; see also the composite analysis in Figs. S5 and S6). The red "spur" in quadrant IV of the scatter plot becomes more evident as the sample size increases. During the summers between consecutive La Niña events, the AAC index exhibits a positive shift compared with the expectation based on the preceding winter ENSO alone. Once we include the concurrent summer ENSO, the scatter is narrowed towards the diagonal (Fig. 2d; the partial regression coefficients are both significant at 1% using a Student's $t$-test), and the correlation between the reconstructed and simulated AAC indices increases from 0.3 to 0.48 ($p < 0.001$ for this

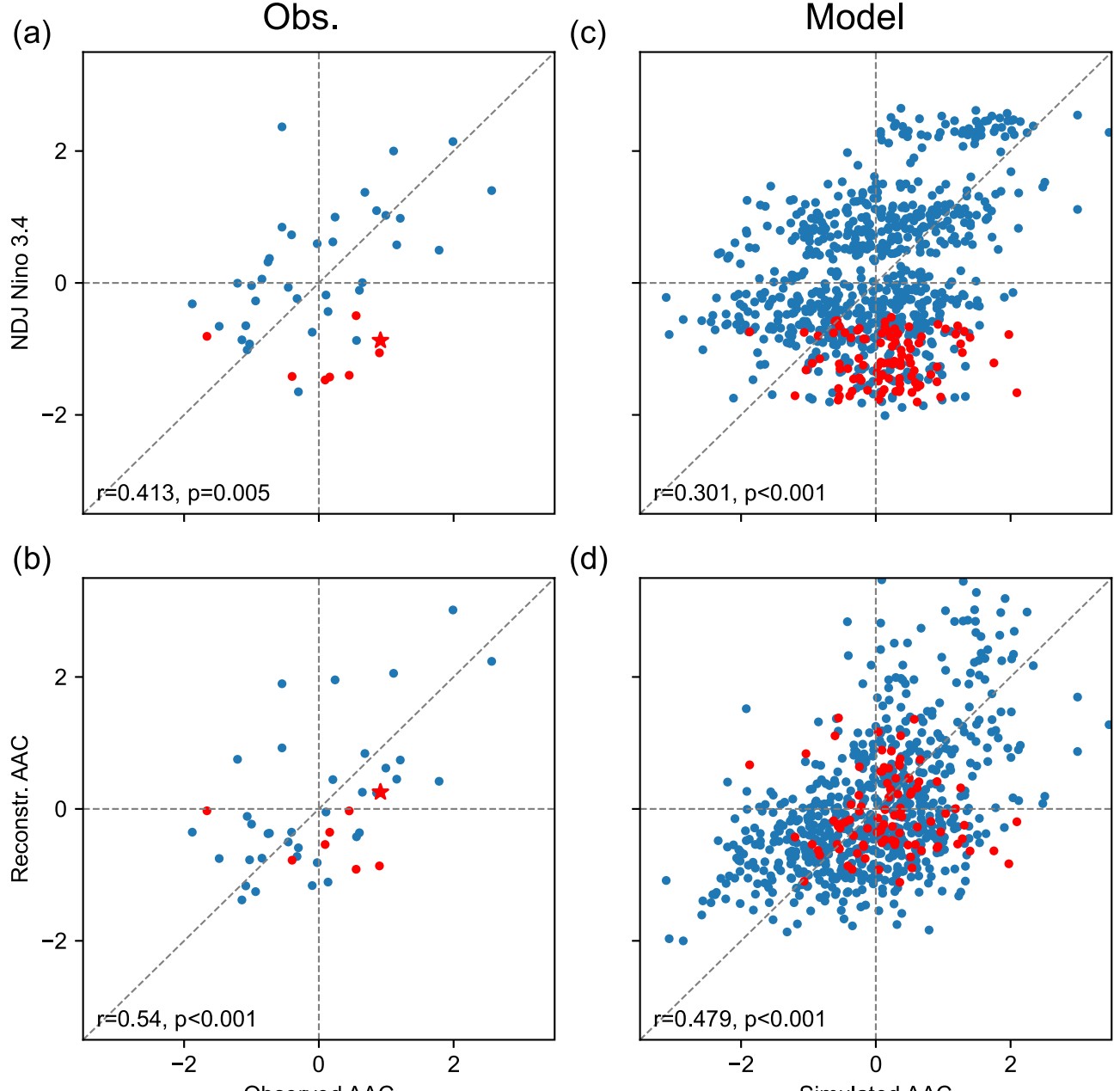

**Fig. 2 | Scatterplots showing the asymmetry of summer anomalous anticyclone (AAC). a** Observed AAC index (x-axis) versus preceding winter ENSO (y-axis). **b** Observed AAC index (x-axis) versus reconstructed AAC by preceding winter and concurrent summer ENSO states (y-axis). **c**, **d** Same as (**a**, **b**) but for all individual ensemble members in model simulations. All the indices are normalized. The red dots denote summers between consecutive La Niña, and the red star in (**a**, **b**) denotes the summer of 2022. Source data are provided as a Source Data file.

change), accompanied by an 8% reduction in the root mean square error. This confirms the substantial impact of summer ENSO, particularly La Niña, on AAC events.

The preceding winter ENSO and the ensuing summer AAC are linked by the capacitor effect of the Indian Ocean, where ENSO-induced Indian Ocean anomalies persist through the post-ENSO summer due in part to slow-propagating oceanic downwelling Rossby waves[10,12,38]. As a proxy of the Indian Ocean capacitor effect, the North Indian Ocean (NIO; 0–20°N, 50–120°E) SST anomalies in summer correlates better with AAC than the preceding winter ENSO (Fig. S8; cf. Fig. 2). Incorporating summer ENSO significantly improves the prediction based on JJA NIO SST in model simulations as well ($p < 0.001$), further confirming the role of concurrent summer ENSO in modulating AAC variability, especially during consecutive La Niña events.

### Effects of preceding and concurrent ENSO

Recognizing the distinct role of consecutive La Niña, we quantify the specific impacts of the preceding and simultaneous ENSO on the summer AAC. We apply a similar multiple linear regression model (See details in "Methods" section) to SLP, sea surface temperature (SST), precipitation, and 850 hPa winds. The influence of preceding El Niño events exhibits a typical AAC pattern in the NWP, south of which the easterly wind anomalies extend over the north IO to warm the SST by decelerating the prevailing monsoonal westerlies (Fig. 3a). Meanwhile, warm SST anomalies and enhanced precipitation in the IO excite an atmospheric equatorial Kelvin wave to support the AAC via low-level Ekman divergence, resulting in a positive interbasin feedback (Fig. 3b, c)[39–41]. The meridional alignment of the SLP disturbances appears as a high-low-high tripole over the Philippines, Japan, and the

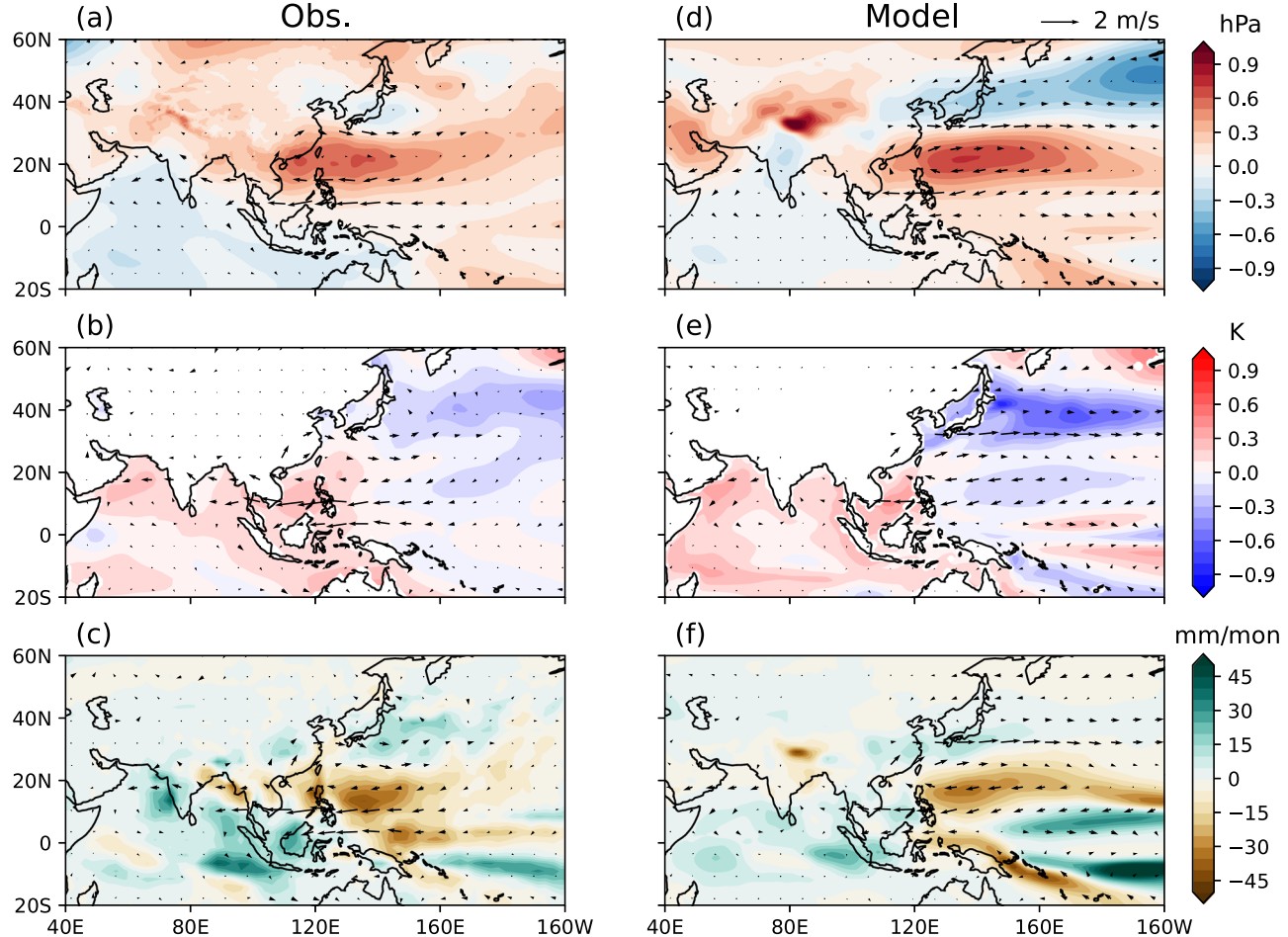

**Fig. 3 | Contributions from preceding winter El Niño. a–c** Impacts of preceding El Niño conditions on summer sea level pressure (SLP), sea surface temperature (SST), precipitation (shadings in three panels respectively), and 850 hPa winds (arrows in all panels) represented by the partial regression coefficients against normalized El Niño-Southern Oscillation (ENSO) index in preceding winter in observations. **d–f** Same as (**a–c**), but for model simulations. Source data are provided as a Source Data file.

Okhotsk Sea, similar to the Pacific-Japan (PJ) pattern (Fig. 3a)[42,43]. The CGCM generally captures the AAC pattern with a similar extent and location, as well as the associated large-scale SST and precipitation anomalies in the NWP and IO, though the modeled anticyclone does not stretch as far west as in observations (Fig. 3d–f). Since the CGCM simulation is by design only constrained by the preceding ENSO, the results confirm these features as coupled responses to El Niño conditions in the preceding winter.

We move next to the contribution from the summer ENSO. For ease of comparison, the coefficients obtained from the regression model are multiplied by −1 to represent the impact of simultaneous La Niña. In the equatorial region, the negative Indian Ocean Dipole and La Niña are distinct from the post-El Niño anomaly pattern. The observed SLP response features broader anomalies representing La Niña teleconnections (Fig. 4a), and low SST suppresses precipitation in the equatorial Pacific (Fig. 4b, c). High-pressure anomalies can be identified in the NWP through the South China Sea to the Bay of Bengal, which projects onto the AAC, albeit displaced slightly southeast compared with its counterpart forced by the preceding El Niño (cf. Fig. 3).

The large ensemble of CGCM simulations confirms the contribution from simultaneous La Niña events by capturing the high-pressure response that projects onto the AAC in the NWP (Fig. 4d–f). The location and extent of the SLP anomaly resemble the post-El Niño counterparts in observations (Fig. 4d; cf. Fig. 3a). The associated easterly wind anomalies reach the north IO, consistent with those observed. Despite overall more intense precipitation over the equatorial Pacific and the Maritime Continent, the dry anomaly in the NWP and the wet anomaly in the tropical IO linked to the AAC are well represented (Fig. 4f).

## Implications for forecasts

Climate anomalies induced by preceding and concurrent ENSO both project onto the AAC pattern. Perfect prediction of summer Niño 4 SST increases the skill of the multi-regression reconstruction of AAC to $r = 0.54$, from $r = 0.41$ based only on the preceding winter Niño 3.4 (Fig. 2a, b). The ensemble mean of the CGCM hindcast achieves a skill of $r = 0.48$ in predicting the observed AAC (Fig. 5a), in part because the model predicts JJA Niño 4 SST at $r = 0.57$ and is capable of simulating consecutive La Niña (e.g., see Fig. S7c, d). When the model is initialized on May 1st from the pacemaker experiment, the skill in predicting JJA Niño 4 SST increases to 0.77 (Fig. S9; $p = 0.05$ for this change), and the simulation bias of summer AAC is alleviated in the right direction in seven out of eight consecutive La Niña events. Therefore, accurate prediction of summer ENSO contributes to a successful seasonal forecast for the EASM, much as for the South Asian summer monsoon[44].

The observed AAC is a mixture of coupled atmosphere-ocean feedback and atmospheric stochastic variability, and the latter limits the skill of seasonal prediction. Our 20-member initialized CGCM hindcast yields 20 possible outcomes for summer AACs, providing an estimate of the uncertainty range (Fig. 5c). In most years, the observed

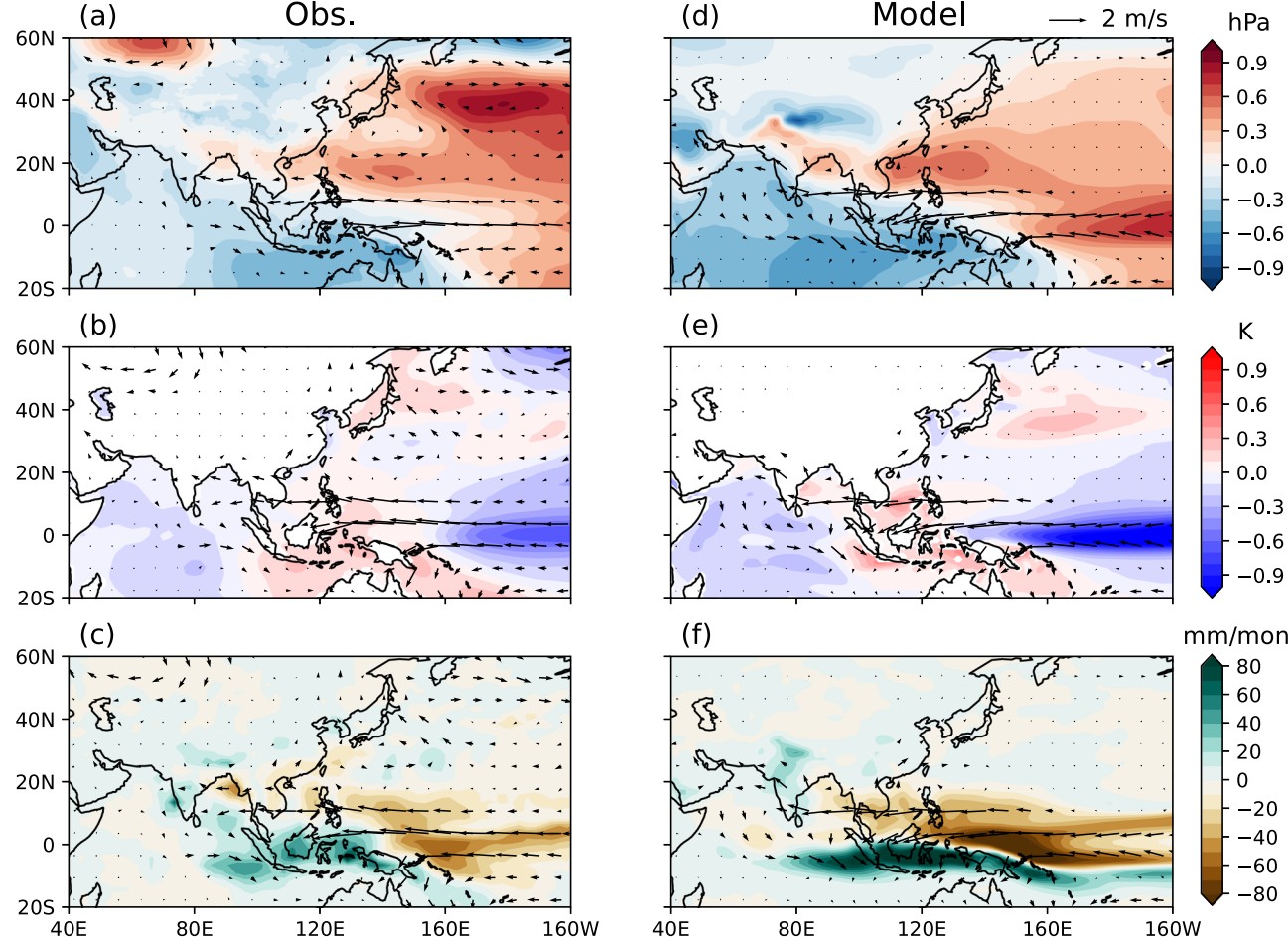

**Fig. 4 | Contributions from concurrent summer La Niña. a–c** Impacts of concurrent La Niña events on sea level pressure (SLP), sea surface temperature (SST), precipitation (shadings in three panels respectively), and 850 hPa winds (arrows in all panels) represented by the partial regression coefficients (signs flipped) against normalized El Niño-Southern Oscillation (ENSO) index in concurrent summer in observations. **d–f** Same as (**a–c**), but for model simulations. Source data are provided as a Source Data file.

AAC indeed lies within the range of 20 realizations. In 2022, the ensemble-mean Niño 4 SST anomaly is nearly zero for JJA (Fig. S7) but there are a few members that capture the summer La Niña and the summer AAC (see run #4 as an example in Fig. S10). The ensemble mean of the CGCM hindcast suppresses atmospheric internal variability, providing an upper limit of predictability. With a perfect prediction of summer Niño 4 SST, the multi-regression reconstruction yields a skill as high as $r = 0.84$ against the simulated ACC variability (Fig. 5b), as opposed to $r = 0.60$ when only the preceding winter ENSO is used as a predictor.

## Discussion

The low-level anomalous anticyclone over the NWP is the leading recurrent pattern of interannual variability of the East Asian summer monsoon. We have investigated an apparent asymmetry in the post-ENSO capacitor effect on the NWP AAC, i.e., EASM anomalies are more robust in post-El Niño than post-La Niña summers. ENSO is asymmetric itself: El Niño typically lasts for one year while La Niña often lasts 2–3 years. We show that this ENSO asymmetry gives rise to the asymmetry in post-ENSO AACs. A multi-regression reconstruction based on ENSO phases in the preceding winter as well as the concurrent summer can largely remove the apparent asymmetry. By performing a large-ensemble hindcast initialized on the first day of each January, we obtained 12 times more multi-year La Nina events than in observations. The model results with greatly increased sample size corroborate our conclusions.

Our results provide an explanation for the AAC event of 2022, which was not anticipated based on the preceding winter ENSO condition alone. More generally, the reconstructed AAC variability correlates better with observed or simulated AAC than when only the preceding ENSO condition is considered. This further illustrates the importance of summer ENSO prediction. Accurate forecasts of summer rainfall issued in March/April are crucial for effective preparation for flooding[45]. This in turn requires overcoming the so-called "spring prediction barrier"[46,47] for ENSO prediction, say by improving the representation of stochastic forcing of ENSO, in and outside the tropical Pacific basin, such as by the Pacific meridional mode[25,48–50] and by tropical North Atlantic SST variability[51,52].

The CGCM is capable of reproducing key responses to preceding and concurrent ENSO in the NWP region, but the model has some limitations. For instance, it does not fully capture precipitation anomalies on small scales (see the difference between observations and the model over southern China in Figs. 3 and 4). This may be due to factors such as topography, which the coarse-resolution model does not fully resolve. While these limitations do not undermine our arguments regarding large-scale dynamics, weather/climate forecasts will benefit from continued improvements in models.

## Methods
### Observational datasets and model simulations
The monthly wind and SLP are taken from the latest European Centre for Medium-Range Weather Forecasts (ECMWF) Reanalysis (ERA5)[53].

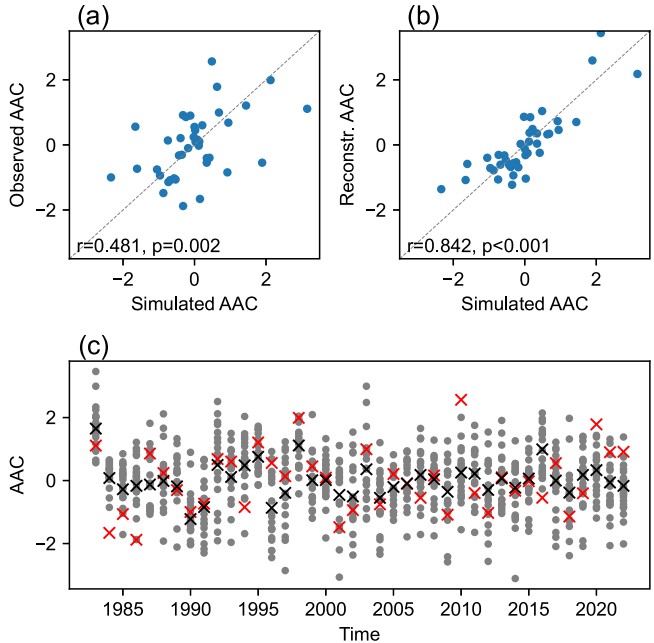

**Fig. 5 | Prediction of summer anomalous anticyclone (AAC). a** AAC index in ensemble mean of coupled general circulation model (CGCM) hindcast (x-axis) versus in observation (y-axis). **b** Simulated AAC index (x-axis) versus reconstructed AAC index with preceding and concurrent El Niño-Southern Oscillation (ENSO) states (y-axis) from the ensemble mean of CGCM hindcast. **c** The summer AAC indices in all ensemble members (gray dots) and in ensemble mean (black crosses) of model simulations, compared with the observed AAC index (red crosses). All the indices in panels (**a**, **b**) are normalized for an easy comparison. Source data are provided as a Source Data file.

The monthly SST is from Version 5 of the Extended Reconstructed Sea Surface Temperature (ERSST) dataset[54], and the monthly precipitation is from the Global Precipitation Climatology Project (GPCP) Monthly Analysis Product[55]. The data used in this study are from 1979 to 2022 when reliable satellite-based observations become available. Monthly anomalies are obtained by subtracting long-term climatologies from the detrended data.

To further test our hypotheses and enlarge the sample size, we perform a similar analysis on ensemble simulations of a CGCM fulfilled by the Geophysical Fluid Dynamics Laboratory coupled model, version 2.1 (CM2.1, which is capable of reproducing the observed asymmetry in duration between El Niño and La Niña)[56,57] and initialized from the tropical Pacific pacemaker experiment. In the pacemaker experiment, SST anomalies in the tropical Pacific are restored towards observation; the CGCM is then initialized on January 1st of each year and runs freely for another year. Therefore, the CGCM simulation carries ENSO memory from the previous winter while maintaining ocean-atmosphere interactions. The restoring region of the pacemaker experiment and a schematic of the CGCM setup are shown in Fig. 1 of ref. 28. The CGCM simulations cover a period of 1983–2022, and the ensemble size is 20, an order of magnitude larger than the observations for a robust analysis.

## ENSO indices
We use the NDJ average of SST anomalies in the Niño 3.4 region to track the ENSO phase in its peak season. In observations and model simulations constrained by observed SST, El Niño and La Niña thresholds are set as ±0.5 K, and we follow this criterion to identify El Niño and La Niña events. However, in unconstrained simulations, since models tend to simulate ENSO with higher variance, we change the thresholds to ±0.5 standard deviation. Consecutive La Niña events are

therefore identified as La Niña threshold being reached in two or three consecutive winters.

We use the JJA average of the SST anomalies in the Niño 4 region to track the phase of summer ENSO. It is argued that SST in Niño 4 region can better determine La Niña episodes, as a −0.5 K SST anomaly in Niño 4 is sufficient to bring SST below the 28-degree threshold, resulting in a substantial zonal shift of deep convection[35]. Geographic proximity of the Niño 4 region allows a direct influence of Niño 4 SST anomalies on the AAC. We indeed see that JJA SST in the Niño 4 region can modulate AAC more efficiently than in the Niño 3.4 region.

## Identification of the AAC mode
Following ref. 28, we define an AAC index as the first principal component (PC1) of the leading EOF of JJA-mean SLP anomalies in the NWP region (0–35°N, 100°–160°E; purple box in Fig. 1b), and the AAC mode is represented as the regressions of SLP and wind against the AAC index (Fig. 1b).

Previous studies using indices based on wind shear[58] or vorticity[59] in a similar domain captured a similar atmosphere-ocean coupling pattern. An exception might be ref. 60, which uses a much larger domain and includes more variability over the North Pacific, a region that is not the focus of our research. Otherwise, our analysis in this study is insensitive to the choice of indices.

## Multiple linear regression
We use multiple linear regression to separate and quantify the relative contributions of ENSO in the preceding winter and the concurrent summer. The independent (predictor) variables are NDJ Niño 3.4 SST and JJA Niño 4 SST, respectively. We are not overly concerned about multicollinearity since these two independent variables are typically uncorrelated, a manifestation of the so-called "spring persistence barrier" (Fig. S11)[61,62].

To reconstruct the AAC index based on ENSO phases in the preceding winter and concurrent summer, we first use multiple linear regression to quantify the relative contributions of ENSO in two seasons to the observed AAC, and then add the ENSO contributions together for a reconstructed AAC index:

$$AAC(t) = \hat{b}_1 \times ENSO_{NDJ}(t) + \hat{b}_2 \times ENSO_{JJA}(t) + \hat{a} + \varepsilon(t), \quad (1)$$

$$AAC_{reconstr.}(t) = \hat{b}_1 \times ENSO_{NDJ}(t) + \hat{b}_2 \times ENSO_{JJA}(t), \quad (2)$$

where $ENSO_{NDJ}$ and $ENSO_{JJA}$ represent indices for the preceding winter and concurrent summer.

To specify the ENSO's contribution to a specific variable, the regression model is generalized as

$$X(t,x,y) = \hat{b}_1(x,y) \times ENSO_{NDJ}(t) + \hat{b}_2(x,y) \times ENSO_{JJA}(t) + \hat{a}(x,y) + \varepsilon(t,x,y), \quad (3)$$

where X is an arbitrary variable of interest. The respective contribution is represented by the field of regression coefficients $\hat{b}_1(x,y)$ and $\hat{b}_2(x,y)$. $ENSO_{NDJ}$ and $ENSO_{JJA}$ are standardized so that $\hat{b}_1(x,y)$ and $\hat{b}_2(x,y)$ can be compared directly.

## Tests of statistical significance
We use Fisher's Z-transformation test[63,64] to test the significance of a change in correlation. Unless otherwise stated, all other significance tests are based on Student's t-test.

## Data availability
The ERA5 data are available at the Copernicus Climate Data Store (https://doi.org/10.24381/cds.6860a573). The ERSST dataset is available at https://doi.org/10.7289/V5T72FNM. The GPCP dataset is

available at https://psl.noaa.gov/data/gridded/data.gpcp.html. The raw CGCM simulations are available from the corresponding author upon request. Source data underlying the main figures are provided with this paper. Source data are provided with this paper.

## Code availability

The scripts for generating the figures are available at https://doi.org/10.5281/zenodo.13308862[65].

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

## Acknowledgements
P.Z., S.-P.X., and A.M. are supported by the National Science Foundation (AGS 2105654). Y.K. is supported by the Japanese Ministry of Education, Culture, Sports, Science and Technology (JPMXD0722680395) and by the Japan Society for the Promotion of Science (JP19H05703, JP23K22573, JP23K25937, JP23K25946, and JP24H02223). N.J.L. is supported by the National Science Foundation (2023520). Y.M.O. is supported by the National Science Foundation (AGS 2105641).

## Author contributions
Conceptualization: P.Z., S.-P.X. Methodology: P.Z., S.-P.X., Y.K., A.M. Investigation: P.Z., S.-P.X. Visualization: P.Z. Supervision: S.-P.X., N.J.L., Y.M.O. Writing—original draft: P.Z., S.-P.X. Writing—review & editing: P.Z., S.-P.X., Y.K., N.J.L., Y.M.O., A.M.

## Competing interests
The authors declare no competing interests.
