## [Peer Review File · Nature Communications]

Why East Asian Monsoon Anomalies Are More Robust in Post El Niño than in Post La Niña SummersREVIEWER COMMENTS

Reviewer #1 (Remarks to the Author):

The AAC is the most important circulation pattern affecting the interannual variation of EASM variability, and it is well known that the AAC appears usually in the summers following the preceding winter El Niño. Therefore, the preceding winter El Niño has been considering as an important prediction factor and this factor is widely used to predict summer climate in East Asian countries, with the most successful example of 1998 summer. However, there are also many cases of extreme AAC appearance in the summers without preceding winter El Niño and even in the summers such as 2022 with preceding winter La Niña, making the seasonal prediction of East Asian summer climate complicated and lack of reliability. It has been believed that the unique feature of internal atmospheric variability in the WNP and East Asia is the main cause of complicated relationship between the AAC and preceding winter El Niño, and thus the predictability of East Asian summer climate would be limited and hard to be improved.

This study suggested that consecutive La Niña events can affect the atmospheric circulation in summer over the WNP in a quite different way compared with single La Niña events. That is, in summer between consecutive La Niña events, AAC tends to appear, in a sharp contrast to the anomalous cyclone in summer following single La Niña events in the preceding winter. Using a large ensemble experiment makes this conclusion reliable. The present results have important implications for seasonal prediction of East Asian summer climate, i.e., the consecutive La Niña events should be distinguished from single La Niña events.

The paper would draw wide attention, and has been well written. I recommend accept after some revisions suggested as follows.

1. It seems to be that the contrast between consecutive and single La Niña events would be more scientifically interesting and significant, compared with the asymmetry between El Niño and La Niña. So, comparison with single La Niña events is recommended. In the current version, there are not results on single La Niña events.
2. It would be more straightforward to show the composite anomalies for preceding winter El Niño, consecutive (and single) La Niña events, compared with partial regression coefficients. If the authors agree, please replace the regression results by composite results. If not, please state the advantage of partial regression analysis.
3. Lines 176-182, including Supplementary text and Fig. S3: I don't think this discussion on the summer preference is closely related the main topic of this study. In addition, the AAC and associated anomalous easterlies in tropical western Pacific can be well explained by the SST and precipitation anomalies, so the analysis of energy conversion seems not to be necessary.

Reviewer #2 (Remarks to the Author):

Reviewer #3 (Remarks to the Author):

Review of 'Why are East Asian monsoon anomalies more robust in post-El Nino than post-La Nina summers?' by of Zhang et al., submitted to Nature Communications

General comments:

In this paper the authors demonstrate that the evolution of ENSO affects the North West Pacific anticyclone in boreal summer (June to August), the strength of which has been linked to key climate phenomena such as heatwaves and heavy rainfall in eastern Asia. In particular, while the state of ENSO in the previous winter (when ENSO phases typically peak in strength) is well known to influence these climatic elements, it is shown here that whether ENSO remains in the same phase or transitions to the opposite phase following winter is also important in determining the response in this region. This result is inferred from historical observations but is then confirmed using climate model simulations that offer a much larger sample size.

Overall, this is a nice result which helps rationalise to some degree the apparent diversity of responses to similar ENSO phases. I think it should make a good contribution to the literature. Nevertheless, the authors need to do more to demonstrate the robustness of their results (see comments below) and give a fuller exposition - in places the discussion is too cursory. I also think that the authors restrict themselves too much in the use of observational data - a larger dataset could reasonably be obtained to help address the sampling issues.

A further element that I think is missing is connecting the prior (winter) phase with the response. We can do this statistically of course, but physically there have to be steps in the mechanism to allow a 6-month delayed response. This is commonly accepted to occur via secondary SST anomalies in the Indian Ocean and West Pacific (similar to those in fig. 3b). So an obvious thing to ask is whether in some cases the establishment of these anomalies fails or is weaker than normal, thereby leading to the expected response not materialising. I think it would be fairly simple to test whether a springtime index of this aspect in place of winter Nino 3.4 made an improvement. This would be very useful for the reader in getting a fuller picture of what is going on.

The paper is well written and overall is well illustrated. Some figures have panels which are too

small, however, or have features which are not clear (see specific comments below). I strongly recommend these aspects are improved before acceptance.

Specific comments:

39. The 'framework' suggested here is not realistic. With a correlation of 0.4, no one would imagine that there should be an inevitable progression from a given ENSO phase to a specific sign of anomaly in NWP MSLP. I suggest that the text here is rewritten to avoid this artificial construct.

47. I suggest it would be useful to add a timeseries of the AAC index to fig.1, highlighting the post-El Nino and post-La Nina years, to make this point clearer for the reader.

55. Note also that El Nino conditions can be persistent on occasion. Perhaps the correct way to express this is that there are more examples of persistent La Nina in historical records than persistent El Nino.

59. The statement about model capabilities is very sweeping and not particularly helpful in my view. It is also rather strange given that the authors go on to use model analysis in this paper to provide a 'better' estimate of the effect of ENSO influences than is available from observations. In fact, some models show relatively realistic ENSO variability. Of course, there are many facets of the phenomenon and it is bound to be possible to demonstrate that some of these are deficient in any given model. The relevant thing here is whether models are deficient in the aspects that control this particular teleconnection, but no evidence either way is provided on this question. I suggest the authors remove or reformulate this point.

61. 'substantially complicates' is somewhat opaque. What does this mean? Is it intended to mean that it adds noise? If so, I suggest that the authors say this directly.

78. I suggest referring to fig. S1 once more here.

79. Presumably this difference in correlations is statistically significant, but this should be properly demonstrated.

82. Where 'mostly' is used here it would be better to provide the numbers. The text would benefit from greater precision here.

85. It is not clear how the content of the sentence beginning 'Therefore, ...' follows from that of the previous sentence. Please clarify the text here.

94. Please add a reference or references to justify the statement that La Nina decays more slowly than El Nino. Also, isn't it a single, protracted event rather than consecutive events?

102. Please add a reference or references to justify the statement about moderate SST perturbations in Nino 3 having little atmospheric effect.

109. Is this change in correlation significant? Please evaluate this and report it here.

113. This statement about data reliability is too pessimistic. Certainly, SST data for the Tropical Pacific are available going back further, and it is not necessary to use a compound index for ENSO. There are time-dependent uncertainties in SST, but these are not vastly larger in the mid-20th century than in the modern day (see e.g. Kennedy et al. 2014). I would estimate that data from the 1950s onwards is suitable, as this time reflects a rise in ship observations across the Pacific. Regardless of this, however, modern SST datasets are ensembles of realisations reflecting the full range of uncertainties in their creation, so confidence can easily be evaluated.

Kennedy, J. J. (2014), A review of uncertainty in in situ measurements and data sets of sea surface temperature, *Rev. Geophys.*, 52, 1–32, doi:10.1002/2013RG000434.

115. '...', the Geophysical Fluid Dynamics ...!

131. These results are definitely suggestive of an improvement, but there needs to be more done here to demonstrate this in a quantitative way, i.e. establishing that it is statistically significant. Given that the results are computed using MLR models, I wonder whether RMS error is a sensible addition to the change in correlation? I guess overall I feel that the discussion of these results is too cursory in the current manuscript.

144-147. There is definitely a first-order comparability between the model and obs in terms of pattern. However, there are also some differences, particularly for China (which motivates this study). It appears that the anticyclone does not stretch as far west as in observations and there is not a corresponding dry signal over Southern China. These differences are not mentioned, yet they seem important to me if the spatial patterns are to be analysed alongside large-scale indices. Some further discussion of fig. 3 is necessary in my view.

172-174. Again, some mention of the shortcomings of the simulations would give the reader a better all round view of the results. I notice once more that the rainfall signals tend to fall short of mainland China.

191. These simulations are initialised in January, so are hardly 'seasonal' forecasts in the normal sense. A 'true' seasonal forecast, initialised at the beginning of May, should show somewhat better ENSO skill.

223. I contend that it is not 'puzzling'. This relates to my earlier comment that with moderate predictive skill from the prior (winter) ENSO phase, we would not be surprised to see responses of the opposite sign to those expected from this relationship. The event would just be attributed to atmospherically generated 'noise'. What this study shows is that the continuation of La Nina in this year helped to modify these expectations. I note, however, that the reconstructed anomaly using both the prior and concurrent ENSO states (fig. 1b) is still smaller than observed, so perhaps there was an element of noise involved as well.

259. How do the results change using an in-season (i.e. JJA) measure of ENSO is used?

261. These are not especially compelling arguments to use Nino 4 in my view - is proximity a better motivation?

275. I would avoid saying that this study allowed 'too much' variance from other parts of the NP. It sounds like a value judgement on the study in question, which I am sure the authors don't mean to make.

Multiple figures: the wind arrows are too difficult to see well in several figures. Please increase the size and additionally thin them in necessary.

Figure 2. The panels are too small to be very clear. Please increase the size by at least 50%. Additionally, it would be helpful to add a fit line to the La Nina to La Nina years to show how they do/do not conform to expectations based on all data.

Figures 3 and 4. It would be preferable, I think, to show stippling for significance instead of the wind vectors.

Supplementary information:

34. Please provide references to the derivation of this equation.

52. Change to 'crosses'.

Fig. S5. Would be good to increase the panel size as previously suggested for some of the other figures.

Response to Reviewers

We thank the reviewers for their thorough reading of the manuscript and for their helpful and constructive comments. The manuscript has been much improved by addressing their comments.

The reviewers' comments are listed below in *blue italics* and our responses in regular text. Unless otherwise stated, line numbers and page numbers refer to the revised manuscript.

Reviewer #1:

The AAC is the most important circulation pattern affecting the interannual variation of EASM variability, and it is well known that the AAC appears usually in the summers following the preceding winter El Niño. Therefore, the preceding winter El Niño has been considering as an important prediction factor and this factor is widely used to predict summer climate in East Asian countries, with the most successful example of 1998 summer. However, there are also many cases of extreme AAC appearance in the summers without preceding winter El Niño and even in the summers such as 2022 with preceding winter La Niña, making the seasonal prediction of East Asian summer climate complicated and lack of reliability. It has been believed that the unique feature of internal atmospheric variability in the WNP and East Asia is the main cause of complicated relationship between the AAC and preceding winter El Niño, and thus the predictability of East Asian summer climate would be limited and hard to be improved.

This study suggested that consecutive La Niña events can affect the atmospheric circulation in summer over the WNP in a quite different way compared with single La Niña events. That is, in summer between consecutive La Niña events, AAC tends to appear, in a sharp contrast to the anomalous cyclone in summer following single La Niña events in the preceding winter. Using a large ensemble experiment makes this conclusion reliable. The present results have important implications for seasonal prediction of East Asian summer climate, i.e., the consecutive La Niña events should be distinguished from single La Niña events.

The paper would draw wide attention, and has been well written. I recommend accept after some revisions suggested as follows.

We thank the reviewer for the thorough reading and thoughtful comments. Our point-by-point reply follows.

1. It seems to be that the contrast between consecutive and single La Niña events would be more scientifically interesting and significant, compared with the asymmetry between El Niño and La Niña. So, comparison with single La Niña events is recommended. In the current version, there are not results on single La Niña events.

Following the suggestion, the revision compares consecutive and single-year La Niña events at L88-91:

“The composite analysis based on observational data reveals that the summer monsoon anomalies between consecutive La Niña events do not exhibit a typical negative AAC pattern, in contrast to those observed in summers following single-year La Niña events (Fig. S4).”

Fig. S4. Composites of observed post-La Niña summer anomalies. (a to c) SLP, SST, precipitation (shading in three panels respectively), and 850 hPa winds (arrows in all panels) averaged in post single-year La Niña summers. **(d to f)** Same as **(a to c)** but for summers between consecutive La Niña events. The two composites each contain 8 individual cases.

We decided to place the composite figures (Figs. S3-6) into supplementary materials, for the reasons explained in our response to comment #2 below.

2. It would be more straightforward to show the composite anomalies for preceding winter

El Niño, consecutive (and single) La Niña events, compared with partial regression coefficients. If the authors agree, please replace the regression results by composite results. If not, please state the advantage of partial regression analysis.

Following the suggestion, we have included composite analysis to reinforce our arguments. We prefer to place the composite results (Figs. S3-6) in the supplementary materials and keep the partial regression results in the main text for the following reasons:

- 1) The composite analysis does not make full use of available data. For example, to show the sole effects of La Niña in the concurrent summer, we would need to subtract the “single-year La Niña” composite from the “consecutive La Niña” composite. Each composite, based on the observational record, contains fewer than ten events. In contrast, partial regression analysis utilizes all the available data.
- 2) Quantitative comparison in composite analysis is difficult. In the example above, the average strengths of single-year or consecutive La Niña events are not necessarily the same. We would need to rescale the composites before subtracting one from another. Similarly, the average strengths of El Niño and La Niña are different, complicating quantitative comparisons.

3. Lines 176-182, including Supplementary text and Fig. S3: I don't think this discussion on the summer preference is closely related the main topic of this study. In addition, the AAC and associated anomalous easterlies in tropical western Pacific can be well explained by the SST and precipitation anomalies, so the analysis of energy conversion seems not to be necessary.

Following the suggestion, we have deleted this part to keep the manuscript concise.

Reviewer #2:

We thank the reviewer for the constructive review.

Reviewer #3:

Review of 'Why are East Asian monsoon anomalies more robust in post-El Nino than post-La Nina summers?' by of Zhang et al., submitted to Nature Communications

General comments:

In this paper the authors demonstrate that the evolution of ENSO affects the North West Pacific anticyclone in boreal summer (June to August), the strength of which has been linked to key climate phenomena such as heatwaves and heavy rainfall in eastern Asia. In particular, while the state of ENSO in the previous winter (when ENSO phases typically peak in strength) is well known to influence these climatic elements, it is shown here that whether ENSO remains in the same phase or transitions to the opposite phase following winter is also important in determining the response in this region. This result is inferred from historical observations but is then confirmed using climate model simulations that offer a much larger sample size.

Overall, this is a nice result which helps rationalise to some degree the apparent diversity of responses to similar ENSO phases. I think it should make a good contribution to the literature. Nevertheless, the authors need to do more to demonstrate the robustness of their results (see comments below) and give a fuller exposition - in places the discussion is too cursory. I also think that the authors restrict themselves too much in the use of observational data - a larger dataset could reasonably be obtained to help address the sampling issues.

A further element that I think is missing is connecting the prior (winter) phase with the response. We can do this statistically of course, but physically there have to be steps in the mechanism to allow a 6-month delayed response. This is commonly accepted to occur via secondary SST anomalies in the Indian Ocean and West Pacific (similar to those in fig. 3b). So an obvious thing to ask is whether in some cases the establishment of these anomalies fails or is weaker than normal, thereby leading to the expected response not materialising. I think it would be fairly simple to test whether a springtime index of this aspect in place of winter Nino 3.4 made an improvement. This would be very useful for the reader in getting a fuller picture of what is going on.

The paper is well written and overall is well illustrated. Some figures have panels which are too small, however, or have features which are not clear (see specific comments below). I strongly recommend these aspects are improved before acceptance.

We thank the reviewers for the thorough reading and thoughtful comments. We have provided the results of significance tests to demonstrate the robustness and explained our

reasons to confine the study to the period of post-1979 (see our responses to the specific comments below).

Regarding the comment on secondary SST anomalies, we repeated the analysis of Fig. 2 but replaced preceding winter Niño 3.4 SST with JJA NIO SST anomalies. We indeed found improved correlations compared with Fig. 2 in all the panels. The asymmetry associated with consecutive La Niña events is still present (77% of the red dots are below the diagonal in panel c), though it is less pronounced than in Fig. 2. Incorporating summer ENSO significantly increases the correlations ($p < 0.001$ for the change from 0.4 to 0.56 in model simulations), confirming that summer ENSO enhances prediction skills.

Fig. S8. As in Fig. 2 but replacing NDJ Niño 3.4 SST with JJA NIO SST. (a) Observed AAC index (x-axis) versus JJA NIO SST anomalies (y-axis). (b) Observed AAC index (x-axis) versus reconstructed AAC by JJA NIO SST and concurrent summer ENSO states (y-axis). (c) and (d): Same as (a) and (b) but for all individual

ensemble members in model simulations. All the indices are normalized. The red dots denote summers during consecutive La Niña.

We added a paragraph at L145-152 regarding these results:

“The preceding winter ENSO and the ensuing summer AAC are linked by the capacitor effect of the Indian Ocean, where ENSO-induced Indian Ocean anomalies persist through the post-ENSO summer due in part to slow-propagating oceanic downwelling Rossby waves^{10,12,38}. As a proxy of the Indian Ocean capacitor effect, the North Indian Ocean (NIO; 0-20°N, 50-120°E) SST anomalies in summer correlates better with AAC than the preceding winter ENSO (Fig. S8; cf. Fig. 2). Incorporating summer ENSO significantly improves the prediction based on JJA NIO SST in model simulations as well ($p < 0.001$), further confirming the role of concurrent summer ENSO in modulating AAC variability, especially during consecutive La Niña events.”

We decided to place the figure in the supplementary materials, because the secondary SST response, though important in the AAC dynamics, is not the main focus of our present study.

Our replies to specific comments are listed below.

Specific comments:

39. The 'framework' suggested here is not realistic. With a correlation of 0.4, no one would imagine that there should be an inevitable progression from a given ENSO phase to a specific sign of anomaly in NWP MSLP. I suggest that the text here is rewritten to avoid this artificial construct.

We rewrote the text to emphasize the probabilistic nature of this relationship at L39-42:

“However, this relationship does not guarantee a deterministic progression; for instance, although an anomalous cyclone (AC, or negative AAC) was probabilistically favored to follow the La Niña in the winter of 2021/2022, the observed outcome was a strong AAC responsible for the record-breaking heatwave.”

47. I suggest it would be useful to add a timeseries of the AAC index to fig.1, highlighting the post-El Nino and post-La Nina years, to make this point clearer for the reader.

We included a figure (Fig. S2) of AAC index time series and highlighted post-El Nino and post-La Nina summers with different colors. Also, we indicated the mean AAC index in the two categories in the figure (L80-81).

55. Note also that El Nino conditions can be persistent on occasion. Perhaps the correct way to express this is that there are more examples of persistent La Nina in historical records than persistent El Nino.

We modified our expression at L57-58:

“ENSO itself is not symmetric: in historical records, consecutive La Niña events are more likely to occur than consecutive El Niño events.”

59. The statement about model capabilities is very sweeping and not particularly helpful in my view. It is also rather strange given that the authors go on to use model analysis in this paper to provide a 'better' estimate of the effect of ENSO influences than is available from observations. In fact, some models show relatively realistic ENSO variability. Of course, there are many facets of the phenomenon and it is bound to be possible to demonstrate that some of these are deficient in any given model. The relevant thing here is whether models are deficient in the aspects that control this particular teleconnection, but no evidence either way is provided on this question. I suggest the authors remove or reformulate this point.

Thanks. We decided to remove this point to avoid any confusion.

61. 'substantially complicates' is somewhat opaque. What does this mean? Is it intended to mean that it adds noise? If so, I suggest that the authors say this directly.

The leading mode of non-ENSO variability also projects onto the AAC and can emerge spontaneously without ENSO forcing (see Zhang et al. 2024, cited in the text). We do not call it “noise” because the non-ENSO AAC has its coupled dynamics and can provide additional predictability for the summer monsoon variability; therefore, we believe it’s not a simple signal-to-noise issue. Nonetheless, this non-ENSO variability is irrelevant to the ENSO-AAC relationship and makes it difficult to extract a clear ENSO-AAC relationship from limited observational data.

To avoid confusion, we modified our text at L61-63:

“Furthermore, the leading mode of non-ENSO variability also projects onto the AAC and can emerge spontaneously, making it difficult to extract a clear ENSO-AAC relationship”

78. I suggest referring to fig. S1 once more here.

Added as suggested.

79. Presumably this difference in correlations is statistically significant, but this should be properly demonstrated.

We performed a significance test and found this difference is marginally significant ($p=0.076$), because the sample size in the observational record is small. Nevertheless, such a small p-value suggests the need for further investigation and highlights the critical role of large-sample-size model simulations. We reported this value and modified the text at L82-84:

“Despite the small sample size, this difference in the magnitudes of the correlations is close to the significance threshold ($p=0.076$) using Welch’s t-test, suggesting the need for further exploration.”

82. Where 'mostly' is used here it would be better to provide the numbers. The text would

benefit from greater precision here.

We modified the text and provided a precise number at L86:

“...whereas 73% of post-El Niño events are confined to quadrant I”.

85. It is not clear how the content of the sentence beginning 'Therefore, ...' follows from that of the previous sentence. Please clarify the text here.

To enhance the logic flow, we explicitly discussed the differences between consecutive and single year La Niña events based on the composite analysis suggested by Reviewer #1 at L88-91:

“The composite analysis based on observational data reveals that the summer monsoon anomalies between consecutive La Niña events do not exhibit a typical negative AAC pattern, in contrast to those observed in summers following single-year La Niña events (Fig. S4). Therefore, consecutive La Niña events emerge as a key factor in resolving the asymmetry.”

94. Please add a reference or references to justify the statement that La Nina decays more slowly than El Nino. Also, isn't it a single, protracted event rather than consecutive events?

We added the reference at L100.

We call them consecutive events because we can still observe the spring decay and autumn re-intensification between two peak winters, rather than a prolonged single event. We reworded the following sentence at L100-102:

“In consecutive La Niña events, cold anomalies in the tropical Pacific can persist throughout the year despite SST anomalies weakening between the two peak winters.”

102. Please add a reference or references to justify the statement about moderate SST perturbations in Nino 3 having little atmospheric effect.

We added two references at L110 on convection sensitivity to SST and seasonal variation of convective SST threshold.

109. Is this change in correlation significant? Please evaluate this and report it here.

Unfortunately, the change in correlation is not statistically significant at 95% confidence level ($p=0.23$) due to the relatively short observational record, even though we indeed see the asymmetry is alleviated in the scatter plots. We add an explanation at L117-118

“Though this change is not statistically significant due to the small sample size ($p=0.23$), the asymmetry in the scatter distribution is alleviated.”

This underscores the importance of model simulations in expanding the sample size: when the sample size increases to 800, the change in correlation from 0.3 to 0.48 is significant ($p<0.001$). We report this value at L141; see our response to the below comment.

113. This statement about data reliability is too pessimistic. Certainly, SST data for the Tropical Pacific are available going back further, and it is not necessary to use a compound

index for ENSO. There are time-dependent uncertainties in SST, but these are not vastly larger in the mid-20th century than in the modern day (see e.g. Kennedy et al. 2014). I would estimate that data from the 1950s onwards is suitable, as this time reflects a rise in ship observations across the Pacific. Regardless of this, however, modern SST datasets are ensembles of realisations reflecting the full range of uncertainties in their creation, so confidence can easily be evaluated.

Kennedy, J. J. (2014), A review of uncertainty in in situ measurements and data sets of sea surface temperature, Rev. Geophys., 52, 1–32, doi:10.1002/2013RG000434.

We agree that reliable SST data in the Pacific can be traced further back. However, in addition to SST, we are also interested in atmospheric states, including pressure, winds, and precipitation. Since atmospheric observations over open oceans are limited prior to 1979, we have focused on the period from 1979 onwards. We have modified our expression accordingly at L121-122 and cited the suggested paper.

“Although reliable SST observations date back to the 1950s, high-quality atmospheric data with global coverage only became available during the satellite era that nominally started in 1979.”

115. '..., the Geophysical Fluid Dynamics ...'!

Corrected.

131. These results are definitely suggestive of an improvement, but there needs to be more done here to demonstrate this in a quantitative way, i.e. establishing that it is statistically significant. Given that the results are computed using MLR models, I wonder whether RMS error is a sensible addition to the change in correlation? I guess overall I feel that the discussion of these results is too cursory in the current manuscript.

We evaluated and reported the statistical significance and calculated the reduction of RMS error. The text has been modified at L140-142:

“...the correlation between the reconstructed and simulated AAC indices increases from 0.3 to 0.48 ($p < 0.001$), accompanied by an 8% reduction in the root mean square error.”

144-147. There is definitely a first-order comparability between the model and obs in terms of pattern. However, there are also some differences, particularly for China (which motivates this study). It appears that the anticyclone does not stretch as far west as in observations and there is not a corresponding dry signal over Southern China. These differences are not mentioned, yet they seem important to me if the spatial patterns are to be analysed alongside large-scale indices. Some further discussion of fig. 3 is necessary in my view.

Following the suggestion, we mentioned the model biases at L164-167:

“The CGCM generally captures the AAC pattern with a similar extent and location, as well as the associated large-scale SST and precipitation anomalies in the NWP and IO,

though the modeled anticyclone does not stretch as far west as in observations (Fig. 3, d to f).”

We also add a paragraph in Discussion section to discuss model shortcomings:

“The CGCM is capable of reproducing key responses to preceding and concurrent ENSO in the NWP region, but the model has some limitations. For instance, it does not fully capture precipitation anomalies on small scales (see the difference between observations and the model over southern China in Figs. 3 and 4). This may be due to factors such as topography, which the coarse-resolution model does not fully resolve. While these limitations do not undermine our arguments regarding large-scale dynamics, weather/climate forecasts will benefit from continued improvements in models.”

172-174. Again, some mention of the shortcomings of the simulations would give the reader a better all round view of the results. I notice once more that the rainfall signals tend to fall short of mainland China.

We mentioned the rainfall differences in southern China in the above-mentioned paragraph in Discussion section. As we are focusing on large-scale features, we prefer to place the discussion on smaller-scale rainfall disparities in the following section, keeping the main results section concise.

191. These simulations are initialised in January, so are hardly 'seasonal' forecasts in the normal sense. A 'true' seasonal forecast, initialised at the beginning of May, should show somewhat better ENSO skill.

Following the suggestion, we conducted an additional set of simulations that are initialized from the Pacific pacemaker simulations on May 1st each year and indeed found a better prediction skill for JJA Niño 4 SST, with r increasing from 0.57 to 0.77 ($p=0.05$). This increase does not translate to a statistically significant improvement in AAC prediction skill, probably because we only have a 40-year-long record to compare. Nevertheless, we checked the performance of AAC in consecutive La Niña events and found that in seven out of eight events, the simulation bias of summer AAC is alleviated in the right direction in the seasonal forecast initialized in May. We briefly reported the results of May-initialized seasonal forecast at L204-207:

“When the model is initialized on May 1st from the POGA pacemaker run, the skill in predicting JJA Niño 4 SST increases to 0.77 (Fig. S9; $p=0.05$ for this change), although this improvement is not sufficient to yield a statistically significant increase in AAC prediction skill over the relatively short observational record.”

223. I contend that it is not 'puzzling'. This relates to my earlier comment that with moderate predictive skill from the prior (winter) ENSO phase, we would not be surprised to see responses of the opposite sign to those expected from this relationship. The event would just be attributed to atmospherically generated 'noise'. What this study shows is that

the continuation of La Nina in this year helped to modify these expectations. I note, however, that the reconstructed anomaly using both the prior and concurrent ENSO states (fig. 1b) is still smaller than observed, so perhaps there was an element of noise involved as well.

We have rewritten the text to be more objective at L240-241

“Our results provide an explanation for the AAC event of 2022, which was not anticipated based on the preceding winter ENSO condition alone.”

259. How do the results change using an in-season (i.e. JJA) measure of ENSO is used?

The selection of consecutive La Niña events is intended to suggest that these events differ from other ENSO events in inducing the summer AAC response (red dots in Fig. 2). The subsequent quantitative results from the partial regression analysis do not rely on this selection criterion.

We examined the summer Niño 3.4 SST anomalies for the selected consecutive La Niña events. In observations, Niño 3.4 SST anomalies in seven out of the eight events are near or below -0.5K, with the exception of summer 2017. In model simulations, all consecutive La Niña events show strong cold signals in summer (below -0.5 standard deviation). Therefore, we believe the qualitative picture remains consistent.

261. These are not especially compelling arguments to use Niño 4 in my view - is proximity a better motivation?

In response to the comment, we have reworded the text at L290-291:

“Geographic proximity of the Niño 4 region allows a direct influence of Niño 4 SST anomalies on the AAC.”

The arguments for using Niño 4 SST are widely accepted. For example, NOAA presents similar arguments regarding Niño regions (<https://www.ncei.noaa.gov/access/monitoring/enso/sst>). We would like to retain the arguments.

275. I would avoid saying that this study allowed 'too much' variance from other parts of the NP. It sounds like a value judgement on the study in question, which I am sure the authors don't mean to make.

In response to the comment, we modified the expression at L300-302:

“An exception might be ref. 59, which uses a much larger domain and includes more variability over the North Pacific, a region that is not the focus of our research.”

Multiple figures: the wind arrows are too difficult to see well in several figures. Please increase the size and additionally thin them in necessary.

We have modified the wind arrows in the concerned figures to improve visibility.

Figure 2. The panels are too small to be very clear. Please increase the size by at least 50%. Additionally, it would be helpful to add a fit line to the La Nina to La Nina years to show

how they do/do not conform to expectations based on all data.

Following the suggestion, we have modified Fig. 2 to improve readability. We also attempted to add a fit line for consecutive La Niña events (the red dots), but found it did not add much clarity. Our point is that the red dots systematically deviate from the diagonal, which can be effectively shown by different colors. Therefore, we prefer to keep the scatter plot as it is.

Figures 3 and 4. It would be preferable, I think, to show stippling for significance instead of the wind vectors.

We prefer to keep the wind vectors in these figures, as the AAC is a coupled mode in which wind anomalies play an important role in the inter-basin positive feedback. Instead, we evaluate statistical significance of our reconstructed AAC index. For both observations and model simulations, the partial regression coefficients are all statistically significant ($p < 0.01$). We report the results in the revision (L104-107 and L138-140).

“We use a multiple linear regression model to consider the NDJ Niño 3.4 SST anomalies, as well as June-August (JJA) SST anomalies in the Niño 4 region to track the concurrent ENSO (see details in Methods; the partial regression coefficients are both significant at 1% using a Student’s t-test), ...”

“Once we include the concurrent summer ENSO, the scatter is narrowed towards the diagonal (Fig. 2d; the partial regression coefficients are both significant at 1% using a Student’s t-test)...”

Supplementary information:

34. Please provide references to the derivation of this equation.

This equation was used in Maloney and Hartmann (2001) and Kosaka and Nakamura (2010). Following reviewer #1’s suggestion, we decided to delete the analysis on energy conversion to keep the manuscript concise and to the point.

52. Change to 'crosses'.

Corrected.

Fig. S5. Would be good to increase the panel size as previously suggested for some of the other figures.

We increased the panel size of this figure.

References

Zhang, P., Xie, S.-P., Kosaka, Y. & Lutsko, N. J. Non-ENSO Precursors for Northwestern Pacific Summer Monsoon Variability with Implications for Predictability. *J. Clim.* 37,

199–212 (2024).

Maloney, E. D. & Hartmann, D. L. The Madden–Julian Oscillation, Barotropic Dynamics, and North Pacific Tropical Cyclone Formation. Part I: Observations. *J. Atmospheric Sci.* 58, 2545–2558 (2001).

Kosaka, Y. & Nakamura, H. Mechanisms of meridional teleconnection observed between a summer monsoon system and a subtropical anticyclone. Part I: The Pacific–Japan pattern. *J. Clim.* 23, 5085–5108 (2010).

Wang, C.-Y., Xie, S.-P. & Kosaka, Y. Indo-Western Pacific Climate Variability: ENSO Forcing and Internal Dynamics in a Tropical Pacific Pacemaker Simulation. *J. Clim.* 31, 10123–10139 (2018).

REVIEWERS' COMMENTS

Reviewer #1 (Remarks to the Author):

The authors have revised the manuscript according to our review comments, and responded reasonably. The co-reviewer and I are satisfied with the revisions made by the authors, and have no further suggestions for possible improvement. Thus, we recommend accept.

Reviewer #2 (Remarks to the Author):

Reviewer #3 (Remarks to the Author):

Second Review of 'Why are East Asian monsoon anomalies more robust in post-El Nino than post-La Nina summers?' by of Zhang et al., submitted to Nature Communications

I commend the authors for making substantial improvements to the manuscript following my first review comments. The paper now seems very good and is certainly suitable for publication. Although I have offered a few additional comments below in the hope these might help the authors, I do not require to see a further revision.

The confirmation of the Indian Ocean link using summer IO SST as a predictor of AAC is a good addition. It offers more of a sense of seasonal prediction potential than the January-initialised simulations. It also makes identification of the role of summer ENSO even more clear.

I accept the explanation that the limitation on the observational period comes from the atmospheric data rather than the SST. Nevertheless, I would urge the authors consider how far back analyses can be pushed more carefully in future work. It is easy to make the default assumption that the cut-off in the reliability of observational data over the Pacific is the start of the satellite era. But there were in situ observations before this, and whether earlier reanalysis is too poor for any given study is not obvious. If we (as a community) complain that we struggle with statistical inference due to lack of length of record (as was commented in this paper), then I think we need to make sure that where we choose to cut off is close to where the true limits of the information content of historical data lie. This is just an optimisation problem balancing improved statistics

from a longer record length with worse statistics due to greater data uncertainty. The great thing is that nowadays we have ensemble reanalyses that allow the effect of these earlier uncertainties to be assessed and the optimal length to be identified.

General comments:

123. With the modified justification for the analysis period, the phrasing of this sentence becomes not quite correct. Also, using a model does not 'remedy' the situation, because the model is imperfect and unlikely therefore to be a substitute for a long good-quality observed record. I suggest 'This restriction on the analysis period limits our ability to make robust inferences about the ENSO-AAC relationship, particularly as it reduces the number of multi-year La Nina events available. To address this limitation, we perform ...'.

141. I recommend that it is made clearer here that the p-value refers to the difference in correlations, following the comment in the previous review.

208. This conclusion looks a little odd given that the previous sentence states that evidence for this is lacking (for the reason clearly stated). Maybe if some more of the justification/findings that were included in the authors' responses were included in the manuscript here, this might look more credible.

Response to Reviewer #3

We thank the reviewers for re-reviewing our manuscript and providing further comments. The comments of reviewer #3 are listed below in *blue italics* and our responses in regular text. The line numbers refer to the revised manuscript.

Reviewer #3:

General comments:

123. With the modified justification for the analysis period, the phrasing of this sentence becomes not quite correct. Also, using a model does not 'remedy' the situation, because the model is imperfect and unlikely therefore to be a substitute for a long good-quality observed record. I suggest 'This restriction on the analysis period limits our ability to make robust inferences about the ENSO-AAC relationship, particularly as it reduces the number of multi-year La Nina events available. To address this limitation, we perform ...'.

We revised the text as the reviewer suggests at Lines 125-128.

141. I recommend that it is made clearer here that the p-value refers to the difference in correlations, following the comment in the previous review.

We revised the text at Line 146:

“... ($p < 0.001$ for this change).”

208. This conclusion looks a little odd given that the previous sentence states that evidence for this lacking (for the reason clearly stated). Maybe if some more of the justification/findings that were included in the authors' responses were included in the manuscript here, this might look more credible

We revised the text to emphasize the improvement in AAC prediction during multi-year La Nina events at Lines 211-212:

“... and the simulation bias of summer AAC is alleviated in the right direction in seven out of eight consecutive La Niña events.”